# Sparse data imputation with Bayesian non-linear factor analysis

## Abstract

We propose a new method for non-linear modeling of latent variables and factors via random Fourier features for high-dimensional data. Essentially, we apply a basis function expansion of a factor analysis model to approximate a Gaussian process mapping of the latent variable and the latent factors to the observed data space. This paper demonstrates the effectiveness of our proposed model with experiments on real datasets in comparison with competing latent variable models. In particular, we show that our proposed model is effective for missing data imputation, especially when the percentage of missing data is high.

## 1 Introduction

Latent variable models (LVMs) are a commonly used statistical method which models the observed data with a lower dimensional representation. Many popular LVMs model the observed data as a linear combination of the *latent variables*, which are the lower dimensional representation of the data, and the *latent factors*, which are the lower dimensional representation of the observed features. Linear models are popular because they are often convenient to fit from a computational perspective. However, they are limited in terms of their expressivity as a consequence of the linear assumption.

The Gaussian process latent variable model (GPLVM) has been extensively used in all sorts of machine learning tasks such as classification, image recognition and recommendation systems (Lawrence, 2004; Li & Chen, 2016). Gaussian processes (GP) are stochastic processes over functions with a continuous domain (Williams & Rasmussen, 2006). In particular, they offer a Bayesian nonparametric framework for modeling nonlinear latent variable models from observed data when used as a prior over non-linear functions. GPLVMs can be viewed as a multiple-output GP regression model where only the output data are given and the inputs are unobserved and treated as latent variables. The GPLVM assumes that the functional variables are generated by GP from some low-dimensional latent variables that need to be inferred from data. In model inference, we can learn the latent variables by integrating out the functional variables and maximizing the log-marginal likelihood.

Approximate inference techniques in the GPLVM are typically computationally tractable when the observed data likelihood is assumed to be a Gaussian distribution. In this setting, we can obtain closed-form approximations to the log marginal likelihood. However, outside this setting we must rely on some further approximations to this model. One approach is to use a random Fourier feature approximation of the kernel function. To solve the statistical problem with GPLVM in non-Gaussian settings, a recent method random feature latent variable model (RFLVM) is developed by (Gundersen et al., 2021; Zhang et al., 2023). This method uses a random Fourier feature approximation in representation of the GP-distributed function approximation, enabling a computationally feasible inference procedure. It produces closed-form gradients of the posterior regardless of the choice of likelihood which thereby facilitates a Markov chain Monte Carlo sampling procedure for exact Bayesian inference. Moreover, this method applies for various types of observations, such as binomial, multinomial, and negative binomial, which allow us to uncover the latent manifold structure of count data.

In this work, we introduce a novel methodology for leveraging low-dimensional embeddings through a dual latent factor analysis framework.

We extend the idea of random feature latent variable modeling by incorporating a non-linear combination of the latent factors in the data generating process. In particular, we utilize one latent variable to model the manifold of observations and a separate latent variable for the manifold of features. This approach allows us to capture and analyze the complex relationships present in the data more effectively. We call our method the "random feature latent factors analysis" (RFLFA). Our approach here for posterior inference relies on using the elliptical slice sampler (Murray et al., 2010), which is a Markov chain Monte Carlo algorithm for performing posterior inference in parameters with multivariate Gaussian priors and (typically) non-conjugate likelihoods.

Our primary application will focus on the problem of missing data imputation, which arise in almost all applied statistical analysis. One common approach to deal with missingness is to simply discard the observations with missing data. However, it will lead to estimates with larger standard errors due to reduced sample size in most realistic settings (Gelman & Hill, 2006). In addition, one can impute missing values based on the observed data (mean for example) or fill in the last value carried forward, but it can be advantageous for bias (Gelman & Hill, 2006). Our model can produce results comparable with other competing imputation methods in terms of the prediction error on a diverse range of data types, including movie ratings, images, and RNA-sequencing data.

This paper is structured as follows. Section 2 provides the notation and backgrounds. In Section 3 we describe the setting of our model and then introduce the sampling steps. In Section 4, experiments with different missing percentages and latent dimensions are carried out on simulated data and several real datasets. And we also explore its application. In the end, we conclude in Section 5.

## 2 Background Work

In a typical latent variable model, we have some observed data $\mathbf{Y} \in \mathbb{R}^{N \times J}$ where $N$ is the number of observations and $J$ is the dimension of the observed features. Each observation is associated with a latent variable, $\mathbf{X} \in \mathbb{R}^{N \times D}$, such that the latent dimensionality is $D \ll J$. A typical assumption in an LVM is that the relationship between the observed data and the latent variables is linear:

$$\mathbf{y}_i = \mathbf{x}_i \cdot \mathbf{Q} + \epsilon_i, \tag{1}$$

given a projection matrix (sometimes called a factor loading), $\mathbf{Q} \in \mathbb{R}^{D \times J}$, and a noise term, $\epsilon_i$. This formulation of the LVM unifies similar concepts, like factor analysis, collaborative filtering, principal component analysis and matrix factorization (Roweis & Ghahramani, 1999). Principal component analysis can be viewed as an LVM under a noiseless assumption, where the latent variables are an orthogonal projection of the data into lower dimensions Pearson (1901). Spearman (1904) first introduced the concept of a factor analysis model in a social scientific setting, where psychologists are interested in measuring traits, such as intelligence, that are not directly observable from the data. Later, Bayesian variants of factor analysis and matrix factorization have been introduced, which assume Gaussian priors on latent variables and factor loadings (Mnih & Salakhutdinov, 2007; Press & Shigemasu, 1989). Using our notation, the data generating process for these linear factor models is:

$$\mathbf{x}_i \sim \mathcal{N}_D(0, \sigma_X^2 I), \ \mathbf{q}_d \sim \mathcal{N}_J(0, \sigma_Q^2 I), \ \mathbf{y}_i \sim \mathcal{N}_J(\mathbf{x}_i \cdot \mathbf{Q}, \sigma^2 I). \tag{2}$$

Posterior inference for this type of model is not difficult to implement, due to the linear combination of the latent variable and the projection matrix as well as the conjugate relationship between the priors and the likelihood (Roweis, 1997; Tipping & Bishop, 1999; Bishop, 1999). However, the linear assumption becomes overly restrictive if we believe the data generating process is something more complicated than a linear model.

## 2.1 Gaussian Process Latent Variable Models

To depart from the basic linear assumption in Equation 1, we may instead model the data generating process from the latent space to the observed data using a non-linear function. Specifically, we will assume the mapping function from $\mathbf{X}$ to $\mathbf{Y}$ is generated from a Gaussian process–thereby leading to the GPLVM (Lawrence, 2004). The primary task in modeling data using GPLVMs is to learn the lower dimensional representation, $\mathbf{X}$, of the data, $\mathbf{Y}$. The data generating process is:

$$\mathbf{y}_i \sim \mathcal{N}_J(f(\mathbf{x}_i), \sigma^2 I), f \sim \mathcal{GP}(\mu, \Sigma_\theta),$$

where $f$ is GP distributed function with mean function $\mu$ and covariance function $\Sigma_\theta$ with hyperparameters $\theta$. Typically, the mean function is assumed to be zero. If $\mathbf{Y}$ has a Gaussian likelihood, then we can obtain the marginal likelihood of $p(\mathbf{Y}|\mathbf{X}, \theta)$ by analytically integrating out $f$ in closed form. Although we cannot obtain the posterior of $\mathbf{X}$ directly, we can resort to numerous approximate inference techniques to infer the posterior of $\mathbf{X}$, such as Laplace approximations, Hamiltonian Monte Carlo methods, and variational methods.

The GPLVM has proven to be a useful non-linear latent variable model in a wide variety of applications, like modeling single-cell RNA-seq data, or 3D poses of human figures (Ahmed et al., 2018; Verma & Engelhardt, 2020; Ek et al., 2007). However, the basic GPLVM only models a lower dimensional representation of the observations. In some scenarios, we may also want a latent representation of the observed $J$ features. We can naturally obtain these latent representations of the observed features in linear models. For example, the parameter $\mathbf{Q}$ from the model in Equation 1 can also represent the embedding of the features into $D$ dimensional space. In a model like latent Dirichlet allocation (LDA), this feature embedding represents the distribution of words in a topic and can be interpreted as the content of a particular topic. The feature embedding parameter is particularly desirable in collaborative filtering problems where we have $N$ users and $J$ items, and we may want to suggest new items for a user to consume, given observed ratings other users have given to the items.

Lawrence & Urtasun (2009) first introduced the relationship between non-linear matrix factorization and GPLVMs by showing the relationship between Bayesian matrix factorization models with probabilistic Principal Component Analysis (PPCA) (Tipping & Bishop, 1999) and then showing the fact that PPCA is a special case of the GPLVM, where the kernel function is the linear kernel, $K(\mathbf{X}, \mathbf{X}') = \mathbf{X}\mathbf{X}^T$. However, they do not explicitly model a feature embedding parameter in this matrix factorization method using GPs. Adams et al. (2010) introduced a GP-based model for collaborative filtering that models user and item embeddings and performs posterior inference using MCMC. However, they do not use any scalable approximations for the GP-distributed mapping functions. Therefore, inference is very slow in most practical applications of collaborative filtering. Kim et al. (2016) approximates the previous model using a Tucker decomposition (Tucker, 1966) and perform posterior inference using the MAP estimate. Moreover, each of the aforementioned methods are only applicable for Gaussian likelihoods. However, many applications of non-linear factor analysis involve count data where the Gaussian assumption is inappropriate.

## 2.2 Random Feature Latent Variable Models

To generalize GPLVMs to non-Gaussian likelihoods, we approximate the GP-distributed maps in GPLVMs using the random Fourier feature (RFF) approximation of the kernel function (Rahimi & Recht, 2008). The RFF approximation relies on two theorems: Mercer's theorem and Bochner's theorem. Mercer's theorem states that we can equivalently represent a positive definite kernel function as an inner product of a feature mapping (Mercer, 1909): $K(x, x') = \langle \varphi(x), \varphi(x') \rangle$ where $x, x' \in \mathbb{R}^D$. Bochner's theorem states that any continuous shift-invariant kernel $K(x, x') = K(x - x')$ on $\mathbb{R}^D$ is positive definite if and only if it is the Fourier transform of a non-negative measure $p(w)$, which is guaranteed to be a density with a properly scaled kernel (Bochner, 1959). Let $\varphi(x) = \exp(iw^T x)$ and $\varphi(x)^*$ denote its complex conjugate. We can see that:

$$K(x - x') = \int_{\mathbb{R}^D} p(w) \exp(iw^T(x - x'))dw = \mathbb{E}_{p(w)}\varphi(x)\varphi(x')^*]. \tag{3}$$

Then we can use Monte Carlo integration to approximate the kernel function:

$$K(x, x') \approx \varphi(x)\varphi(x')^*. \tag{4}$$

For $M$ Monte Carlo samples, and we drop the imaginary part for real-valued kernels of $\varphi(x)$, we have

$$\varphi_{\mathbf{W}}(\mathbf{x}) = \sqrt{\frac{2}{M}} \begin{bmatrix} \sin(\mathbf{w}_1^T\mathbf{x}) \\ \cos(\mathbf{w}_1^T\mathbf{x}) \\ \vdots \\ \sin(\mathbf{w}_{M/2}^T\mathbf{x}) \\ \cos(\mathbf{w}_{M/2}^T\mathbf{x}) \end{bmatrix}, \quad \mathbf{w}_m \overset{i.i.d.}{\sim} p(\mathbf{w}).$$

We draw $M/2$ samples from $p(w)$ and then equivalently represent a kernel method as a linear model with respect to the basis function projection, $\varphi(x)\beta$ by Mercer's theorem. Using this random projection, we can approximate a GP-distributed function $f(x)$ as

$$f(x) = \varphi_{\mathbf{W}}(\mathbf{x})\beta_X. \tag{5}$$

The randomized approximation of this inner product lets us replace expensive calculations involving the kernel with an M-dimensional inner product, which reduce the computational costs of fitting GP regression models from $O(N^3)$ to $O(NM^2)$ (Hensman et al., 2017).

Gundersen et al. (2021) and Zhang et al. (2023) introduced the "Random Feature Latent Variable Model" (RFLVM) which is a RFF-based approximation of a GPLVM. The data generating process of the RFLVM is:

$$\begin{aligned}
\mathbf{Y}_j &\sim \mathcal{L}(g(\varphi_{\mathbf{W}}(\mathbf{X})\boldsymbol{\beta_j}), \boldsymbol{\theta}), \ \boldsymbol{\theta} \sim p(\boldsymbol{\theta}), \\
\mathbf{x}_i &\sim \mathcal{N}_D(0, I), \mathbf{w}_m \sim \mathcal{N}_D(\boldsymbol{\mu}_{z_m}, \boldsymbol{\Sigma}_{z_m}), \\
z_m &\sim \mathrm{CRP}(\alpha), \alpha \sim \mathrm{Ga}(a_\alpha, b_\alpha), \\
(\boldsymbol{\mu}_k, \boldsymbol{\Sigma}_k) &\sim NIW(\boldsymbol{\mu}_0, \nu_0, \lambda_0, \boldsymbol{\Psi}_0).
\end{aligned} \tag{6}$$

$\mathcal{L}$ here represents the likelihood function, $g$ is an invertible link function that maps $\varphi_{\mathbf{W}}(\mathbf{X})\boldsymbol{\beta_j}$ to the support of the likelihood, and $\theta$ are other likelihood-specific parameters, if they exist (for example, the noise parameter in a Gaussian distribution). As $p(w)$ is a Dirichlet process mixture of Gaussians (DP-GMM, Ferguson (1973); Antoniak (1974)). We assign each $w_m$ in $W = [w_1, \ldots, w_{M/2}]$ to a mixture component with the variable $z_m$, which is distributed according to a Chinese restaurant process (CRP,Aldous (1985)) with concentration parameter $\alpha$. The Dirichlet process is commonly used as a prior for infinite mixture models, where it is assumed that the data comes from a mixture of an infinite number of subpopulations or clusters. In this context, $\alpha$ controls the effective number of clusters by determining the degree of concentration or dispersion in the distribution. This prior introduces additional random variables: the mixture means $\mu_k$, and the mixture covariance matrices $\Sigma_k$ where $K$ is the number of clusters in the current Gibbs sampling iteration.

## 3 Random Feature Latent Factor Analysis

In our random feature latent factor analysis model, we now model the data with a non-linear mapping of both the latent variables and latent factors:

$$
\begin{aligned}
y_{ij} &\sim \mathcal{L}(g(\varphi_{\mathbf{W}}(\mathbf{x}_i) \cdot \boldsymbol{\beta} \cdot \varphi_{\mathbf{W}}(\mathbf{q}_j)^T), \boldsymbol{\theta}), \ \boldsymbol{\theta} \sim p(\boldsymbol{\theta}) \\
\mathbf{X}_i &\sim \mathcal{N}_D(0, I), \mathbf{Q}_j \sim \mathcal{N}_D(0, I), \\
\boldsymbol{\beta} &= \boldsymbol{\beta}_X^T \boldsymbol{\beta}_Q, \ \text{where } \boldsymbol{\beta}_X, \boldsymbol{\beta}_Q \sim \mathcal{N}(0, \Sigma_0), \\
\mathbf{w}_m &\sim \mathcal{N}_D(\boldsymbol{\mu}_{z_m}, \boldsymbol{\Sigma}_{z_m}), \\
z_m &\sim \mathrm{CRP}(\alpha), \alpha \sim \mathrm{Ga}(a_\alpha, b_\alpha), \\
(\boldsymbol{\mu}_k, \boldsymbol{\Sigma}_k) &\sim NIW(\boldsymbol{\mu}_0, \nu_0, \lambda_0, \boldsymbol{\Psi}_0).
\end{aligned}
\tag{7}
$$

In essence, we model a lower-dimensional embedding of the observations using one GP and model another lower-dimensional embedding of the features using another GP.

### 3.1 Posterior Inference

The posterior distribution of the model is intractable to obtain in closed form. Thus, we must resort to a Markov Chain Monte Carlo (MCMC) sampling algorithm to draw samples from the posterior distribution. The initial step involves estimating the posterior of the covariance kernel parameters, denoted as $\{\boldsymbol{w}_m, \boldsymbol{\mu}_k, \boldsymbol{\Sigma}_k, z_m\}$. The covariance kernel is approximated with the RFF, where we assume an infinite Gaussian-inverse Wishart mixture for the frequencies, $\boldsymbol{w}_m$. To explore the posterior of this Dirichlet process mixture, we first sample the latent indicators, $z_m$, which indicate the latent mixture from which each frequency is generated. For this process, we follow Algorithm 8 from Neal (2000), a standard sampling algorithm for DPMMs. Posterior sampling $\boldsymbol{\mu}$ and $\boldsymbol{\Sigma}_k$ relies on established conjugacy results in the context of Gaussian-inverse Wishart mixtures. Next, we apply the Metropolis–Hastings (MH) algorithm, as outlined in Oliva et al. (2016), to sample $W_m$, where the prior distribution on $W_m$ serves as the proposal distribution. Additionally, we sample the concentration parameter $\alpha$ through a variable augmentation scheme to ensure that $\alpha$ remains conditionally conjugate (Escobar & West, 1995).

In this paper, we utilizze the elliptical slice sampler (ESS) to take posterior samples of the latent variables, $\boldsymbol{X}$, and the factor loadings, $\boldsymbol{Q}$ (Murray & Adams, 2010), where the detailed algorithm is presented in Algorithm 2. We opt to use this algorithm for posterior inference because empirical observations suggest that the ESS is more effective for posterior sampling compared to other general MCMC samplers, such as Hamiltonian Monte Carlo or Langevin dynamics MCMC. Murray & Adams (2010) and Gadd et al. (2021) further demonstrate that slice sampling and elliptical slice sampling, respectively, are effective for latent GP models. The linear coefficients $\boldsymbol{\beta}$ are also sampled using ESS, if the associated likelihood does not allow for a closed-form Gibbs sampler. Derivations for the Gibbs sampling step for $\boldsymbol{\beta}$ for the special cases of Gaussian, Poisson and Binomial likelihoods are available in Appendix A.

In our notation, the variables subscripted with zero denote fixed hyperparameters of the model. We initialize all parameters by drawing from the prior distribution, with the exception of $\boldsymbol{X}$ and $\boldsymbol{Q}$, which are initialized using PPCA. This sampling framework facilitates the estimation of the posterior distribution for the latent variables and hyperparameters, enabling robust inference and prediction within the RFLFA. The complete pseudo-code for the posterior inference of our model can be found in Algorithm 1.

## 4 Experiments

Our primary focus in our experimental section is on missing data imputation. In particular, we will demonstrate that our model still attains accurate imputation under a high percentage of missing data compared to standard latent variable models. In these experiments, we calculated the test mean squared error (MSE) by comparing the test set observations, $Y$, with predicted observations $\hat{Y}$ where $\hat{Y}$ is approximated by $\varphi_w(\hat{\mathbf{X}})\beta\varphi_w(\mathbf{Q})^T$. $\hat{\mathbf{X}}$ and $\hat{\mathbf{Q}}$ are the final posterior sample of $\mathbf{X}$ and $\mathbf{Q}$. We held out 20, 40, 60 and 80 percent of the observations for the test set and run each trial three times.

---

**Algorithm 1** Sampling procedure

---

1:  Initialization:
2:    a. Initialize $\boldsymbol{X}$ as the latent variable from PPCA model and $\boldsymbol{Q}$ as the weight matrix
3:    b. Initialize other parameters $\{\boldsymbol{W}, \mu_k, \Sigma_k, \boldsymbol{Z}\}$ from the prior
4:    c. Initialize hyperparameters to be zero
5:  Estimate the posterior of the covariance kernel which involve parameters $\{w_m, \mu_k, \Sigma_k, z_m\}$:
6:    a. Sample $z_m$ following algorithm from Neal (2000):

$$p(z_m = k | \boldsymbol{\mu}, \boldsymbol{\Sigma}, \boldsymbol{W}, \alpha) = \begin{cases} \frac{n_k^{-m}}{M-1+\alpha} \mathcal{N}(\boldsymbol{w}_m | \boldsymbol{\mu}_k, \boldsymbol{\Sigma}_k) & n_k^{-m} > 0 \\ \frac{\alpha}{M-1+\alpha} \int \mathcal{N}(\boldsymbol{w}_m | \boldsymbol{\mu}, \boldsymbol{\Sigma}) \mathrm{NIW}(\boldsymbol{\mu}, \boldsymbol{\Sigma}) d\boldsymbol{\mu} d\boldsymbol{\Sigma} & n_k^{-m} = 0 \end{cases}$$

7:    b. Sample $\mu_k$ and $\Sigma_k$:

$$\boldsymbol{\Sigma}_k \sim \mathcal{W}^{-1}(\boldsymbol{\Phi}_k, v_k), \boldsymbol{\mu}_k \sim \mathcal{N}(\boldsymbol{m}_k, \ \frac{1}{\lambda_k} \boldsymbol{\Sigma}_k)$$

   where
   $\boldsymbol{\Phi}_k = \boldsymbol{\Phi}_0 + \sum (\boldsymbol{w}_m - \bar{\boldsymbol{w}}^{(k)})(\boldsymbol{w}_m - \bar{\boldsymbol{w}}^{(k)})^T + \frac{\lambda_0 n_k}{\lambda_0 n_k} (\boldsymbol{w}_m - \boldsymbol{\mu}_0)(\boldsymbol{w}_m - \boldsymbol{\mu}_0)^T$,
   $\bar{\boldsymbol{w}}^{(k)} = \frac{1}{n_k} \sum_{m:z_m=k}^M \boldsymbol{w}_m, \quad v_k = v_0 + n_k, \quad \boldsymbol{m}^{(k)} = \frac{\lambda_0 \boldsymbol{\mu}_0 + n_k \bar{\boldsymbol{w}}_{(k)}}{\lambda_0 + n_k}, \quad \lambda_k = \lambda_0 + n_k$
8:    c. Sample $\boldsymbol{w}_m$ by Metropolis–Hastings (MH) sampler:

$$\boldsymbol{w}_m^\star \sim q(\boldsymbol{W}) \triangleq p(\boldsymbol{W}|\boldsymbol{z}, \boldsymbol{\mu}, \boldsymbol{\Sigma}), \quad \rho_{\mathrm{MH}} = \min\left\{1, \frac{p(\boldsymbol{Y}|\boldsymbol{X}, \boldsymbol{w}_m^\star, \boldsymbol{\theta})}{p(\boldsymbol{Y}|\boldsymbol{X}, \boldsymbol{w}_m \boldsymbol{\theta})}\right\}$$

9:    d. Sample $\alpha$ by augmenting a variable $\eta$ to make $\alpha$ conditionally conjugate:

$$\eta \sim \mathrm{Beta}(\alpha + 1, M), \quad \frac{\pi_\eta}{1 - \pi_\eta} = \frac{a_\alpha + K - 1}{M(b_\alpha - \log(\eta))}, \quad K = |k : n_k > 0|$$

$$\alpha \sim \pi_\eta \mathrm{Ga}(a_\alpha + K, b_\alpha - \log \eta) + (1 - \pi_\eta)\mathrm{Ga}(a_\alpha + K - 1, b_\alpha - \log \eta)$$

10: Estimate latent variable $\boldsymbol{X}$ and $\boldsymbol{Q}$ iteratively by Elliptical Slice Sampler (refer to algorithm 2).
11: Estimate linear coefficients $\boldsymbol{\beta}$ also by ESS.
12: Gibbs sampling data likelihood-specific parameters from specific distribution.

---

## 4.1 Ovarian Cancer Data

We first investigate the ovarian cancer dataset from Martoglio et al. (2002). This dataset contains 17 tissue samples from 162 genes. We first standardize the data to have zero mean and unit variance. The table below gives the testing MSE of our prediction and the true values. The preceding numbers represent the average values of the three trials, while the numbers in parentheses represent the standard deviation of the three trials.

| Missing Percentage | Latent Dimension | Gaussian RFLVM | Gaussian RFLFA (ours) | GPLVM | PPCA |
|---|---|---|---|---|---|
| 20% | 2 | 0.906 (0.0561) | **0.627** (0.0732) | 0.838 (0.3167) | 0.681 (0.2625) |
| | 3 | 0.939 (0.0241) | **0.727** (0.0427) | 0.896 (0.2554) | 0.744 (0.2294) |
| | 5 | 0.988 (0.0404) | **0.705** (0.0647) | 1.038 (0.2766) | 0.834 (0.2491) |
| | 7 | 0.980 (0.0641) | **0.682** (0.0140) | 1.038 (0.2766) | 0.864 (0.2558) |
| | 9 | 1.003 (0.0353) | **0.645** (0.0300) | 1.038 (0.2766) | 0.833 (0.2575) |
| 40% | 2 | 0.881 (0.0207) | **0.601** (0.0342) | 0.932 (0.1768) | 0.709 (0.1794) |
| | 3 | 0.949 (0.0078) | **0.585** (0.0152) | 0.703 (0.0349) | 0.754 (0.1608) |
| | 5 | 1.003 (0.0195) | **0.506** (0.0029) | 0.932 (0.1768) | 0.793 (0.1664) |
| | 7 | 1.019 (0.0184) | **0.539** (0.0198) | 0.932 (0.1768) | 0.831 (0.1722) |
| | 9 | 0.968 (0.0409) | **0.531** (0.0045) | 0.932 (0.1768) | 0.845 (0.1716) |
| 60% | 2 | 0.953 (0.0221) | **0.463** (0.0545) | 0.922 (0.0989) | 0.916 (0.1109) |
| | 3 | 0.941 (0.0328) | **0.404** (0.0255) | 1.044 (0.1140) | 0.939 (0.1099) |
| | 5 | 0.960 (0.0378) | **0.416** (0.0209) | 1.044 (0.1140) | 0.967 (0.1096) |
| | 7 | 1.005 (0.0361) | **0.422** (0.0335) | 1.044 (0.1140) | 0.983 (0.1104) |
| | 9 | 1.018 (0.0276) | **0.406** (0.0169) | 1.044 (0.1140) | 0.991 (0.1094) |
| 80% | 2 | 0.965 (0.0140) | **0.561** (0.0212) | 0.924 (0.1434) | 0.951 (0.0666) |
| | 3 | 1.022 (0.0102) | **0.562** (0.0234) | 0.992 (0.0657) | 0.960 (0.0612) |
| | 5 | 1.048 (0.0040) | **0.576** (0.0679) | 0.992 (0.0657) | 0.962 (0.0645) |
| | 7 | 1.077 (0.0406) | **0.495** (0.0422) | 0.992 (0.0657) | 0.968 (0.0626) |
| | 9 | 1.111 (0.0196) | **0.469** (0.0262) | 0.992 (0.0657) | 0.971 (0.0636) |

Table 1: Test set MSE with different models on ovarian cancer dataset for four missing percentages. Bold values represent the best results.

Over an array of different missing percentages and latent dimensions, our RFLFA model exhibits the lowest MSE compared to the RFLVM, GPLVM and PPCA. As the missing percentage increases, our model maintains MSE at the same level while the other two models tend to make worse predictions. And our model performs best at 60%, indicating its ability to perform well on sparse data. Furthermore, for each missing percentage, as the latent dimension increases, our model's MSE decreases, whereas the other competing models have a larger MSE.

## 4.2 Wisconsin Breast Cancer Data

The Wisconsin breast cancer data set[1] is another famous cancer data set, which contains 569 instances and 30 features. We also evaluate the performance of our model by comparing predictions with those of other models across various missing data percentages. The Gaussian RFLFA consistently demonstrates lower error rates across almost all missing data scenarios compared to other models, including Gaussian RFLVM, GPLVM, and PPCA. The results suggest that the RFLFA model is particularly robust against varying degrees of missing data. As the percentage of missing values increases (to 40%, 60%, and 80%), RFLFA maintains competitive performance, while other models show a notable increase in error.

---

[1]The data is available at `https://archive.ics.uci.edu/dataset/17/breast+cancer+wisconsin+diagnostic`.

| Missing Percentage | Latent Dimension | Gaussian RFLVM | Gaussian RFLFA (ours) | GPLVM | PPCA |
|---|---|---|---|---|---|
| 20% | 2 | 0.5890 (0.0387) | **0.4547** (0.0330) | 1.0200 (0.0598) | 0.4913 (0.0436) |
| | 3 | 0.6139 (0.0280) | 0.4413 (0.0299) | 1.0200 (0.0598) | **0.4383** (0.0384) |
| | 5 | 0.6743 (0.0493) | 0.4287 (0.0133) | 0.7847 (0.3923) | **0.3743** (0.0384) |
| | 7 | 0.7728 (0.0356) | 0.4677 (0.0500) | 1.0200 (0.0598) | **0.3567** (0.0386) |
| | 9 | 0.8064 (0.0340) | 0.4803 (0.0255) | 1.0200 (0.0598) | **0.3747** (0.0401) |
| 40% | 2 | 0.6922 (0.0301) | **0.4607** (0.0205) | 1.0047 (0.0437) | 0.5720 (0.0348) |
| | 3 | 0.7399 (0.0372) | **0.4573** (0.0258) | 1.0047 (0.0437) | 0.5343 (0.0345) |
| | 5 | 0.7905 (0.0307) | **0.4539** (0.0131) | 1.0047 (0.0437) | 0.4963 (0.0351) |
| | 7 | 0.8418 (0.0226) | **0.4893** (0.0052) | 1.0047 (0.0437) | 0.4953 (0.0380) |
| | 9 | 0.9241 (0.0553) | **0.4850** (0.0332) | 1.0047 (0.0437) | 0.5110 (0.0380) |
| 60% | 2 | 0.8442 (0.0303) | **0.4897** (0.0184) | 0.9970 (0.0290) | 0.7030 (0.0260) |
| | 3 | 0.8826 (0.0285) | **0.4727** (0.0156) | 0.9970 (0.0290) | 0.6797 (0.0251) |
| | 5 | 0.9522 (0.0294) | **0.5000** (0.0112) | 0.7933 (0.3138) | 0.6643 (0.0249) |
| | 7 | 1.0021 (0.0276) | **0.5190** (0.0099) | 0.9970 (0.0290) | 0.6663 (0.0290) |
| | 9 | 1.0460 (0.0356) | **0.5437** (0.0103) | 0.9970 (0.0290) | 0.6783 (0.0266) |
| 80% | 2 | 1.0056 (0.0109) | **0.6363** (0.0161) | 0.9950 (0.0150) | 0.8693 (0.0129) |
| | 3 | 1.0467 (0.0143) | **0.6707** (0.0205) | 0.9950 (0.0150) | 0.8600 (0.0173) |
| | 5 | 1.0942 (0.0167) | **0.6910** (0.0226) | 0.9950 (0.0150) | 0.8593 (0.0189) |
| | 7 | 1.1141 (0.0141) | **0.7323** (0.0251) | 0.9950 (0.0150) | 0.8610 (0.0164) |
| | 9 | 1.1441 (0.0166) | **0.7173** (0.0087) | 0.9950 (0.0150) | 0.8643 (0.0158) |

Table 2: Test set MSE with different models on breast cancer dataset for four missing percentages. Bold values represent the best results.

### 4.3 Spam Data

Next, we wish to demonstrate the performance of our model on a count data set by first analyzing the spam dataset[2] from which we randomly subsampled 2000 observations, set the minimum document frequency of word inclusion in the document-term matrix to a threshold of 1% of documents, and set the maximum threshold of inclusion to 99% of documents leading to 215 words included in the data set.

We compare of RFLFA model with binomial and Poisson likelihoods against PPCA, Poisson factor analysis (PFA) and RFLVM with binomial and Poisson likelihoods. As the missing percentages increase for this particular data set, the binomial RFLFA model performs best in terms of the MSE of the imputed missing data, as seen in Table 3. We can see that almost all the models perform better than the baseline result of predicting all missing values as zero. In addition, we also calculate the test set perplexity on the imputed missing data and further observe that either the Poisson RFLFA or the Binomial RFLFA perform the best.

| Missing Percentage | Latent Dimension | Poisson RFLVM | Poisson RFLFA (ours) | Binomial RFLVM | Binomial RFLFA (ours) | PPCA | PFA | LDA | Baseline |
|---|---|---|---|---|---|---|---|---|---|
| 20% | 2 | 0.0503 (0.0012) | 0.0493 (0.0017) | 0.0517 (0.0029) | **0.0460** (0.0008) | 0.0463 (0.0009) | 0.0473 (0.0017) | 0.0503 (0.0009) | 0.0510 |
| | 3 | 0.0493 (0.0005) | 0.0480 (0.0008) | 0.0567 (0.0025) | **0.0463** (0.0009) | 0.0473 (0.0009) | 0.0717 (0.0132) | 0.0503 (0.0009) | |
| | 5 | 0.0497 (0.0005) | 0.0520 (0.0008) | 0.0757 (0.0024) | **0.0467** (0.0005) | 0.0473 (0.0009) | 0.0637 (0.0087) | 0.0497 (0.0009) | |
| | 7 | 0.0500 (0.0008) | 0.0477 (0.0005) | 0.0887 (0.0031) | **0.0470** (0.0005) | 0.0477 (0.0005) | 0.2463 (0.2062) | 0.0503 (0.0009) | |
| | 9 | 0.0507 (0.0005) | 0.0483 (0.0012) | 0.1017 (0.0012) | **0.0470** (0.0014) | 0.0480 (0.0008) | 0.1537 (0.0477) | 0.0513 (0.0005) | |
| 40% | 2 | 0.0513 (0.0012) | 0.0490 (0.0008) | 0.0560 (0.0008) | **0.0470** (0.0008) | 0.0480 (0.0008) | 0.0480 (0.0008) | 0.0510 (0.0008) | 0.0520 |
| | 3 | 0.0513 (0.0012) | 0.0480 (0) | 0.0623 (0.0039) | **0.0473** (0.0005) | 0.0483 (0.0005) | 0.0617 (0.0052) | 0.0510 (0.0008) | |
| | 5 | 0.0503 (0.0005) | 0.0607 (0.0084) | 0.0890 (0.0022) | **0.0473** (0.0005) | 0.0487 (0.0005) | 0.0623 (0.0046) | 0.0510 (0.0008) | |
| | 7 | 0.0527 (0.0019) | 0.0513 (0.0033) | 0.1080 (0.0071) | **0.0473** (0.0005) | 0.0490 (0.0008) | 0.1513 (0.0882) | 0.0510 (0.0008) | |
| | 9 | 0.0530 (0.0014) | 0.0500 (0) | 0.1190 (0.0033) | **0.0473** (0.0005) | 0.0490 (0.0008) | 0.1186 (0.0264) | 0.0510 (0.0008) | |
| 60% | 2 | 0.0497 (0.0005) | 0.0497 (0.0005) | 0.0587 (0.0008) | **0.0477** (0.0005) | 0.0497 (0.0005) | 0.0480 (0) | 0.0513 (0.0005) | 0.0510 |
| | 3 | 0.0503 (0.0005) | 0.0480 (0.0008) | 0.0697 (0.0012) | **0.0480** (0) | 0.0493 (0.0005) | 0.0606 (0.0034) | 0.0513 (0.0005) | |
| | 5 | 0.0507 (0.0005) | 0.0487 (0.0005) | 0.1023 (0.0049) | **0.0483** (0.0005) | 0.0497 (0.0005) | 0.0593 (0.0041) | 0.0513 (0.0005) | |
| | 7 | 0.0530 (0.0014) | 0.0493 (0.0012) | 0.1273 (0.0031) | **0.0480** (0) | 0.0500 (0.0008) | 0.0820 (0.0042) | 0.0513 (0.0005) | |
| | 9 | 0.0577 (0.0033) | 0.0487 (0.0009) | 0.1460 (0.0065) | **0.0480** (0.0008) | 0.0500 (0) | 0.0927 (0.0068) | 0.0513 (0.0005) | |
| 80% | 2 | 0.0567 (0.0017) | 0.0533 (0.0017) | 0.0690 (0.0008) | **0.0493** (0.0005) | 0.0510 (0) | **0.0493** (0.0005) | 0.0513 (0.0005) | 0.0510 |
| | 3 | 0.0567 (0.0017) | 0.0533 (0.0026) | 0.0837 (0.0031) | **0.0487** (0.0005) | 0.0507 (0.0008) | 0.0547 (0.0017) | 0.0513 (0.0005) | |
| | 5 | 0.0577 (0.0025) | 0.0530 (0.0008) | 0.1277 (0.0031) | **0.0500** (0) | 0.0507 (0.0005) | 0.0563 (0.0037) | 0.0513 (0.0005) | |
| | 7 | 0.0597 (0.0012) | 0.0900 (0.0545) | 0.1563 (0.0031) | **0.0497** (0.0005) | 0.0510 (0) | 0.0643 (0.0033) | 0.0513 (0.0005) | |
| | 9 | 0.0660 (0.0024) | 0.0587 (0.0080) | 0.1727 (0.0147) | **0.0503** (0.0009) | 0.0510 (0) | 0.0737 (0.0037) | 0.0513 (0.0005) | |

Table 3: Test set MSE with different models on spam dataset for four missing percentages. Bold values represent the best results.

---

[2]The spam data set is available here:**https://www.kaggle.com/datasets/uciml/sms-spam-collection-dataset**.

| Missing Percentage | One poisson model | Two poisson model | One binomial model | Two binomial model | PFA | LDA |
|---|---|---|---|---|---|---|
| 20% | 205.26 (11.032) | 154.51 (2.929) | 187.71 (5.564) | **147.38** (3.856) | 189.49 (1.752) | 213.44 (1.437) |
| 40% | 199.02 (7.143) | **153.87** (3.016) | 232.58 (5.404) | 159.30 (5.278) | 190.44 (2.333) | 219.08 (0.437) |
| 60% | 197.25 (7.640) | 156.45 (2.600) | 314.27 (3.187) | **149.22** (3.561) | 206.75 (6.784) | 220.84 (0.268) |
| 80% | 208.31 (6.997) | 168.53 (9.491) | 549.1 (12.972) | **153.31** (4.256) | 275.40 (39.145) | 217.36 (0.327) |

Table 4: Test set perplexity with different models on spam dataset for four missing percentages over 3 trials. Bold values represent the best results. One standard error is reported in parentheses.

## 4.4 CIFAR-10 Data

Next, we looked at the ability for our model to impute missing data in a complicated image data set like CIFAR-10[3]. In our implementation, we first converted the data set to grayscale. PPCA and LDA showed notably higher error rates across all configurations, indicating that traditional statistical methods are less effective in handling the complexities of the CIFAR-10 dataset, particularly in the presence of missing data. Across all missing data scenarios, the Poisson RFLFA consistently achieved lower error rates relative to Poisson RFLVM. For Binomial RFLVM, although the MSE will decrease with increasing latent dimensions, but it behaves badly when the missing percentage is 80%, suggesting potential overfitting. RFLFA models maintained competitive performance even as the missing percentage increased (40%, 60%, 80%).

| Missing Percentage | Latent Dimension | Poisson RFLVM | Poisson RFLFA (ours) | Binomial RFLVM | Binomial RFLFA (ours) | PPCA | LDA |
|---|---|---|---|---|---|---|---|
| 20% | 2 | 13093.66 (19.928) | 2641.96 (379.106) | **1193.78** (59.140) | 2053.21 (46.468) | 2747.49 (6.979) | |
| | 3 | 13305.28 (154.020) | 3440.57 (412.548) | **966.40** (19.612) | 2208.56 (37.523) | 2537.85 (7.180) | |
| | 5 | 13390.498 (142.486) | 3974.97 (756.260) | **815.37** (18.013) | 2428.29 (88.365) | 2382.47 (7.413) | 18441.689 |
| | 7 | 13375.62 (81.231) | 7680.92 (272.061) | **742.63** (2.413) | 2804.93 (332.86) | 2295.24 (7.801) | |
| | 9 | 13670.64 (81.225) | 6085.41 (2326.622) | **665.82** (3.960) | 4040.83 (757.064) | 2242.53 (8.234) | |
| 40% | 2 | 13343.34 (123.364) | 2579.89 (398.233) | 84540.16 (48599.053) | **2110.37** (46.124) | 5206.64 (17.141) | |
| | 3 | 13175.83 (25.168) | 2888.80 (83.026) | 18641.61 (13262.994) | **2381.71** (127.903) | 5092.22 (17.860) | |
| | 5 | 13311.38 (112.316) | 5669.18 (1498.271) | **1281.14** (120.756) | 2924.93 (591.204) | 5048.52 (18.593) | 18444.238 |
| | 7 | 13481.74 (78.169) | 6006.57 (1411.916) | **1008.523** (5.414) | 3148.33 (402.080) | 5044.866 (19.457) | |
| | 9 | 13867.18 (227.063) | 9874.49 (4622.049) | **928.196** (8.848) | 3312.97 (640.410) | 5045.22 (19.457) | |
| 60% | 2 | 13269.91 (118.619) | 2482.00 (22.779) | 86513.38 (65126.258) | **2077.31** (88.920) | 8788.27 (21.387) | |
| | 3 | 13323.46 (26.211) | 2825.22 (210.683) | 19083.79 (15930.015) | **2458.88** (53.635) | 8750.89 (22.311) | |
| | 5 | 13274.53 (210.450) | 2907.32 (376.063) | **1596.96** (259.295) | 2572.20 (284.466) | 8751.25 (22.311) | 18380.378 |
| | 7 | 13862.35 (248.903) | 3837.06 (1008.812) | **1291.66** (87.744) | 2583.99 (191.566) | 8751.60 (22.312) | |
| | 9 | 14432.50 (213.760) | 4755.24 (853.355) | **1087.49** (7.996) | 2687.15 (179.692) | 8751.96 (22.312) | |
| 80% | 2 | 13262.62 (89.987) | 2352.38 (178.327) | 44650.24 (13534.74) | **2154.71** (36.653) | 13286.88 (19.499) | |
| | 3 | 13607.64 (227.756) | 2260.62 (70.652) | 18620.22 (6382.26) | **2250.60** (27.693) | 13283.64 (19.437) | |
| | 5 | 13938.33 (11.492) | 2353.02 (86.284) | 11267.91 (3323.75) | **2365.03** (121.409) | 13284.01 (19.438) | 18397.003 |
| | 7 | 14798.38 (183.244) | 2800.82 (257.421) | 5822.23 (452.787) | **2639.30** (12.382) | 13284.37 (19.439) | |
| | 9 | 15202.25 (204.392) | 2933.53 (514.430) | 5432.85 (738.851) | **2429.91** (129.332) | 13284.73 (19.440) | |

Table 5: Test set MSE with different models on cifar-10 dataset for four missing percentages. Bold values represent the best results.

## 4.5 MovieLens Data

Next, we will look at the application of our model on very sparse data sets, like MovieLens 100k, in order to impute a predicted value for the rating a particular user will give a movie. The data set contains 100,000 ratings from 943 users on 1,682 movies. Even for sparse datasets like MovieLens 100K, we observe that our RFLFA model (and particularly the Binomial RFLFA model) performs the best in terms of imputed MSE, especially over high missing data percentages over all latent dimension sizes. The results show that the proposed method outperforms the original latent variable methods in terms of computational efficiency and accuracy.

---

[3]The CIFAR-10 data is available here: `https://www.cs.toronto.edu/ kriz/cifar.html`

| Missing Percentage | Latent Dimension | Poisson RFLVM | Poisson RFLFA | Binomial RFLVM | Binomial RFLFA | PPCA |
|---|---|---|---|---|---|---|
| | 2 | 3.550 (0.4455) | 1.075 (0.0110) | 1.333 (0.0095) | **0.983** (0.0003) | 8.063 (0.0102) |
| | 3 | 4.775 (1.5585) | 1.143 (0.0662) | 1.379 (0.0713) | **0.980** (0.0045) | 7.640 (0.0086) |
| 20% | 5 | 6.614 (2.7289) | 1.415 (0.0612) | 1.555 (0.1709) | **1.008** (0.0061) | 7.255 (0.0060) |
| | 7 | 20.587 (4.4440) | 1.602 (0.0588) | 1.616 (0.1791) | **1.041** (0.0125) | 7.014 (0.0071) |
| | 9 | 31.480 (3.9281) | 1.799 (0.0593) | 1.615 (0.1278) | **1.055** (0.0224) | 6.895 (0.0070) |
| | 2 | 3.502 (0.4393) | 1.075 (0.0103) | 1.352 (0.0136) | **0.991** (0.0119) | 9.197 (0.0062) |
| | 3 | 4.672 (0.5791) | 1.149 (0.0594) | 1.407 (0.0565) | **0.984** (0.0071) | 8.847 (0.0074) |
| 40% | 5 | 9.133 (1.7232) | 1.406 (0.0163) | 1.578 (0.1722) | **1.021** (0.0108) | 8.554 (0.0075) |
| | 7 | 25.787 (1.2056) | 1.511 (0.1093) | 1.626 (0.1932) | **1.040** (0.0122) | 8.383 (0.0070) |
| | 9 | 28.889 (2.1377) | 1.643 (0.1604) | 1.645 (0.1519) | **1.071** (0.0179) | 8.316 (0.0099) |
| | 2 | 4.323 (0.7443) | 1.135 (0.0225) | 1.371 (0.0151) | **1.011** (0.0128) | 10.526 (0.0052) |
| | 3 | 5.587 (0.2389) | 1.172 (0.0318) | 1.447 (0.0268) | **1.008** (0.0118) | 10.281 (0.0043) |
| 60% | 5 | 25.489 (3.5483) | 1.454 (0.0872) | 1.560 (0.1222) | **1.051** (0.0082) | 10.117 (0.0076) |
| | 7 | 27.782 (0.3972) | 1.512 (0.0645) | 1.636 (0.1624) | **1.074** (0.0302) | 10.039 (0.0069) |
| | 9 | 30.181 (1.6024) | 1.660 (0.0827) | 1.673 (0.1288) | **1.094** (0.0104) | 10.059 (0.0150) |
| | 2 | 8.789 (1.1021) | 1.189 (0.0039) | 1.467 (0.0131) | **1.066** (0.0073) | 12.064 (0.0203) |
| | 3 | 18.347 (0.4822) | 1.296 (0.0232) | 1.509 (0.0364) | **1.084** (0.0063) | 11.954 (0.0222) |
| 80% | 5 | 25.905 (1.9542) | 1.490 (0.0293) | 1.597 (0.1183) | **1.128** (0.0078) | 11.991 (0.0318) |
| | 7 | 31.524 (2.1413) | 1.620 (0.0169) | 1.679 (0.1397) | **1.175** (0.0216) | 12.006 (0.0150) |
| | 9 | 32.482 (4.5758) | 1.639 (0.0861) | 1.683 (0.1106) | **1.167** (0.0186) | 12.032 (0.0186) |

Table 6: Test set MSE with different models on MovieLens 100K dataset for four missing percentages. Bold values represent the best results.

Now, we may further examine some downstream analyses of the results obtained on the MovieLens dataset to demonstrate the utility of our model in a recommendation application. In the first application, we wish to explore the clustering properties of the data with the missing values imputed using RFLFA. Here we use only a subset of the MovieLens 100K data, with users who have ranked more than 100 movies and movies that have been watched by 100 users. Now, the data will have dimensions $267 \times 243$. The cluster analysis, conducted by evaluating the number of clusters ranging from 2 to half the total number of users, indicated that a configuration of 2 clusters yielded the best metric values across all models.

Here, we considered three different metrics to measure the clustering performances without labels–the Silhouette coefficient, Calinski-Harabaz score, and Davies-Bouldin index. The silhouette coefficient measures the relation between the mean intra-cluster distance and the mean nearest-cluster distance. The Calinski-Harabaz score measures how well the ratio of the sum of the between-cluster dispersion and of within-cluster dispersion, thus the larger, the better. The Davies-Bouldin index is also an internal evaluation scheme, with lower values indicating better clustering (Mirkin, 2005). In Table 7 we see that our model yields the best results compared to some simple baseline measures in terms of the clustering metric. Following this, we validated the results of the clustered data by considering the two clustered groups as partitions with distinct behavioral preferences. Here, we treat all the movies they have rated as features to predict the partition that they belong to. The larger absolute value of the coefficient reflects the strength of the feature's influence on the prediction, and therefore we have selected the top fifteen movies modeled by this regression model. It can be seen that our model selects movies that not only overlap with the popular ones but also encompass highly rated and well-known ones.

| Metrics | Mean imputation | Multiple imputation | KNN Imputation | Our model | Based on X | Based on Q |
|---|---|---|---|---|---|---|
| Silhouette Coefficient | 0.0608 | 0.0881 | 0.1212 | **0.4385** | 0.3844 | 0.3581 |
| Calinski-Harabaz Score | 17.5694 | 26.7484 | 37.8887 | **269.0194** | 200.3592 | 155.4608 |
| Davies-Bouldin Index | 3.6245 | 2.9560 | 2.4269 | **0.845** | 0.9848 | 1.0951 |

Table 7: Evaluation on three clustering metrics with different models on a subset of 100K movielense dataset. Bold values represent the best results.

| Methods | Top fifteen movies (in descending order) | | | | |
|---|---|---|---|---|---|
| Most viewed movies | **Star Wars** | **Raiders of the Lost Ark** | Return of the Jedi (Star wars3) | Back to the Future | Empire Strikes Back (Star wars2) |
| | Silence of the Lambs | **Pulp Fiction** | Forrest Gump | Independence Day | **Fargo** |
| | Fugitive | Indiana Jones and the Last Crusade | E.T. | Toy Story | **Monty Python and the Holy Grail** |
| Mean imputation | Citizen Kane(8.3) | Cinderella(7.3) | Home Alone(7.7) | Dead Man Walking(7.5) | Lost World: Jurassic Park(6.5) |
| | Godfather: Part II(9.0) | Leaving Las Vegas(7.5) | Sabrina(6.3) | Big Night(7.3) | Saint(6.2) |
| | Reservoir Dogs(8.3) | Grumpier Old Men(6.6) | To Kill a Mockingbird(8.3) | Maltese Falcon(8.0) | Mask(6.9) |
| Multiple imputation | Grease(7.2) | True Lies(7.3) | Batman Returns(7.1) | Hudsucker Proxy(7.2) | Apocalypse Now(8.4) |
| | Willy Wonka and the Chocolate Factory(7.8) | Jurassic Park(8.2) | Highlander(7.9) | Tin Cup(6.4) | Terminator 2: Judgment Day(8.6) |
| | Star Trek: Generations(6.6) | Apollo 13(7.7) | Searching for Bobby Fischer(7.4) | Sleepless in Seattle(6.8) | Emma(6.6) |
| Knn imputation | Devil's Own(6.2) | Heathers(7.2) | Evita(6.3) | Grumpier Old Men(6.6) | Mighty Aphrodite(7.0) |
| | Blade Runner(8.1) | Casablanca(8.5) | Titanic(7.9) | Sense and Sensibility(7.7) | Lawrence of Arabia(8.3) |
| | Beavis and Butt-head Do America(6.8) | Contact(7.5) | Scream(7.4) | Like Water For Chocolate (7.1) | Net(6.0) |
| Our model | Godfather(9.2) | Schindler's List(9.0) | Usual Suspects(8.5) | Sling Blade(8.0) | Citizen Kane(8.3) |
| | **Raiders of the Lost Ark**(8.4) | To Kill a Mockingbird(8.3) | **Star Wars**(8.6) | Shawshank Redemption(9.3) | **Pulp Fiction**(8.9) |
| | Princess Bride(8.0) | **Fargo**(8.9) | Godfather: Part II(9.0) | Braveheart(8.3) | **Monty Python and the Holy Grail**(8.2) |

Table 8: Movie selection with different methods on a subset of 100K movielense dataset. Bold values represent repeated occured results.

## 5 Conclusion

In this paper, we introduced a non-linear factor analysis model where we push the latent variables and factor loadings through a random Fourier feature basis function–thereby approximating a Gaussian process latent variable model with a lower dimensional projection of both the observed variables and observed features. By using the RFF approximation, we enable a tractable MCMC sampling algorithm to perform posterior inference.

We applied our model for the problem of missing data imputation, and we see in the experimental section that the proposed method can be applied to a wide range of real-world datasets, including image and text processing, as well as natural language understanding. In particular, we see that we obtain the best performance with competing methods when the missing percentage is fairly high. We then looked for a further application in recommendation systems where we typically observe high percentages of missing values. Based on the imputed values in the MovieLens data set, we can carry out some further analyses to identify the most relevant movies to distinguish different clusters of users.

In future work, we are considering extending our proposed method to generalized tensor factorization, like a non-linear Tucker decomposition for example. We believe this will further enhance the expressive capability of GP-based latent variable models to capture higher order non-linear interactions which will further enhance the latent representation of high-dimensional, complex data.

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

# A Derivations for Posterior Sampling

## A.1 Gaussian distribution

If the data likelihood is Gaussian, the model setting is as follows:

$$
\begin{aligned}
\mathbf{Y}_i &\sim \mathcal{N}_j(g(\varphi_{\mathbf{W}}(\mathbf{X})\boldsymbol{\beta}\varphi_{\mathbf{W}}(\mathbf{Q})^T), \Sigma_{\mathbf{Y}}), \ \Sigma_{\mathbf{Y}} \sim \text{Inv-Wishart}(\nu_0, \boldsymbol{\Psi}_0)), \\
\mathbf{X}_i &\sim \mathcal{N}_D(0, I), \mathbf{Q}_j \sim \mathcal{N}_D(0, I), \\
\boldsymbol{\beta} &= \boldsymbol{\beta}_X^T \boldsymbol{\beta}_Q, \ \text{where} \ \boldsymbol{\beta}_X, \boldsymbol{\beta}_Q \sim \mathcal{N}(0, \Sigma_0), \\
\mathbf{w}_m &\sim \mathcal{N}_D(\boldsymbol{\mu}_{z_m}, \boldsymbol{\Sigma}_{z_m}), \\
(\boldsymbol{\mu}_k, \boldsymbol{\Sigma}_k) &\sim NIW(\boldsymbol{\mu}_0, \nu_0, \lambda_0, \boldsymbol{\Psi}_0), z_m \sim \text{CRP}(\alpha), \alpha \sim \text{Ga}(a_\alpha, b_\alpha).
\end{aligned}
$$

The dimension of Y is $N \times J$, dim of X is $N \times D$, dim of Q is $J \times D$. So the dim of $\varphi_{\mathbf{W}}(\mathbf{X})$ is $N \times (M+1)$, dim of $\varphi_{\mathbf{W}}(\mathbf{Q})$ is $J \times (M+1)$ and dim of $\boldsymbol{\beta}$ is $(M+1) \times (M+1)$. We can choose the dimension of $\boldsymbol{\beta}_X$ and $\boldsymbol{\beta}_Q$ to be $M \times (M+1)$. The latent variable $X$ and $Q$ share the same set for $W$.

We have

$$
\begin{aligned}
p(y|\varphi_{\mathbf{W}}(\mathbf{X}), \varphi_{\mathbf{W}}(\mathbf{Q}), \boldsymbol{\beta}, \Sigma_{\mathbf{Y}}) =& (2\pi)^{-\frac{J}{2}} |\Sigma_{\mathbf{Y}}|^{-\frac{1}{2}} \exp(-\frac{1}{2}(y - \varphi_{\mathbf{W}}(\mathbf{X})\boldsymbol{\beta}_X^T(\varphi_{\mathbf{W}}(\mathbf{Q})\boldsymbol{\beta}_Q^T)^T)^T \Sigma_{\mathbf{Y}}^{-1} \\
& \cdot (y - \varphi_{\mathbf{W}}(\mathbf{X})\boldsymbol{\beta}_X^T(\varphi_{\mathbf{W}}(\mathbf{Q})\boldsymbol{\beta}_Q^T)^T)), \\
p(\boldsymbol{\beta}_X) =& (2\pi|\Sigma_0|)^{-\frac{J}{2}} \exp(-\frac{1}{2}\boldsymbol{\beta}_X^T \Sigma_0^{-1} \boldsymbol{\beta}_X), \\
p(\boldsymbol{\beta}_Q) =& (2\pi|\Sigma_0|)^{-\frac{J}{2}} \exp(-\frac{1}{2}\boldsymbol{\beta}_Q^T \Sigma_0^{-1} \boldsymbol{\beta}_Q), \\
p(\Sigma_{\mathbf{Y}}) =& \frac{\Psi_0^{\nu_0}}{2^{\nu_0 p/2} \Gamma_p(\frac{\nu_0}{2})} |\Sigma_{\mathbf{Y}}|^{-(\nu_0+p+1)/2} e^{-\frac{1}{2} tr(\Psi_0 \Sigma_{\mathbf{Y}}^{-1})}.
\end{aligned}
\tag{8}
$$

The posterior may not be a familiar distribution, so we cannot marginalize them out. As the latent variable has Gaussian priors, we can use the elliptical slice sampler to sample them. Similarly, we use elliptical slice sampler to sample $\boldsymbol{\beta}_X, \boldsymbol{\beta}_Q$. And we use Gibbs sampling to sample the likelihood-specific parameter $\Sigma_\mathbf{Y}$.

This is the case that works for real-valued data.

### A.2 Poisson distribution

If the data likelihood is Poisson, the model setting is as follows:

$$
\begin{aligned}
\mathbf{Y}_i \sim & \text{Poisson}(\exp(\varphi_\mathbf{W}(\mathbf{X})\boldsymbol{\beta}\varphi_\mathbf{W}(\mathbf{Q})^T)), \\
\boldsymbol{\beta} = & \boldsymbol{\beta}_X^T \boldsymbol{\beta}_Q, \text{ where } \boldsymbol{\beta}_X, \boldsymbol{\beta}_Q \sim \mathcal{N}(0, \Sigma_0), \\
\mathbf{X}_i \sim & \mathcal{N}_D(0, I), \mathbf{Q}_j \sim \mathcal{N}_D(0, I),
\end{aligned}
$$

$$
L(y_j|\varphi_\mathbf{W}(\mathbf{X}), \varphi_\mathbf{W}(\mathbf{Q}), \boldsymbol{\beta}) = \prod_{i=1}^{N} \frac{1}{y_{ij}!} \frac{\exp(\varphi_\mathbf{W}(\mathbf{X})\boldsymbol{\beta}\varphi_\mathbf{W}(\mathbf{Q})^T)^{y_{ij}}}{e^{\exp(\varphi_\mathbf{W}(\mathbf{X})\boldsymbol{\beta}\varphi_\mathbf{W}(\mathbf{Q})^T)}}. \tag{9}
$$

Similarly, the dimensions for the data and latent variables remain the same. The product of the two latent variables will substitute as the parameter $\lambda$ for the distribution of poisson. Still, we use the elliptical slice sampler to sample $X, Q, \beta_X, \beta_Q$ from the above likelihood, since their priors are still Gaussian.

This is the case that works for integer-valued data.

### A.3 Bernoulli distribution

If the data likelihood is Bernoulli, the model setting is as follows:

$$
\begin{aligned}
\mathbf{Y}_i \sim & \text{Bernoulli}(\text{logistic}(\varphi_\mathbf{W}(\mathbf{X})\boldsymbol{\beta}\varphi_\mathbf{W}(\mathbf{Q})^T)), \\
\boldsymbol{\beta} = & \boldsymbol{\beta}_X^T \boldsymbol{\beta}_Q, \text{ where } \boldsymbol{\beta}_X, \boldsymbol{\beta}_Q \sim \mathcal{N}(0, \Sigma_0), \\
\mathbf{X}_i \sim & \mathcal{N}_D(0, I), \mathbf{Q}_j \sim \mathcal{N}_D(0, I),
\end{aligned}
$$

$$
\begin{aligned}
L(y_j|\varphi_\mathbf{W}(\mathbf{X}), \varphi_\mathbf{W}(\mathbf{Q}), \boldsymbol{\beta}) &= \prod_{i=1}^{N} \left(\frac{\exp(\varphi_\mathbf{W}(\mathbf{X})\boldsymbol{\beta}\varphi_\mathbf{W}(\mathbf{Q})^T))}{1 + \exp(\varphi_\mathbf{W}(\mathbf{X})\boldsymbol{\beta}\varphi_\mathbf{W}(\mathbf{Q})^T)}\right)^{y_{ij}} \left(1 - \frac{\exp(\varphi_\mathbf{W}(\mathbf{X})\boldsymbol{\beta}\varphi_\mathbf{W}(\mathbf{Q})^T))}{1 + \exp(\varphi_\mathbf{W}(\mathbf{X})\boldsymbol{\beta}\varphi_\mathbf{W}(\mathbf{Q})^T)}\right)^{(1-y_{ij})} \\
&= \prod_{i=1}^{N} \frac{(\exp(\varphi_\mathbf{W}(\mathbf{X})\boldsymbol{\beta}\varphi_\mathbf{W}(\mathbf{Q})^T)))^{y_{nj}}}{1 + \exp(\varphi_\mathbf{W}(\mathbf{X})\boldsymbol{\beta}\varphi_\mathbf{W}(\mathbf{Q})^T)}. \tag{10}
\end{aligned}
$$

We use the elliptical slice sampler with above likelihood to sample $X, Q, \beta_X, \beta_Q$ again.
This is the case that works for 0-1 data.

### A.4 Count data distributions

For the binomial, negative binomial distributions, the augmented likelihood has a similar form:

$$
\begin{aligned}
\mathcal{L}(\varphi_\mathbf{W}(\mathbf{X}), &\varphi_\mathbf{W}(\mathbf{Q}), \boldsymbol{\beta}, a(\mathbf{Y}_i), b(\mathbf{Y}_i), c(\mathbf{Y}_i)) \\
&= \prod_{i=1}^{N} c(y_{ij}) \frac{(\exp(\varphi_\mathbf{W}(\mathbf{X})\boldsymbol{\beta}\varphi_\mathbf{W}(\mathbf{Q})^T))^{a(y_{ij})}}{(1 + \exp(\varphi_\mathbf{W}(\mathbf{X})\boldsymbol{\beta}\varphi_\mathbf{W}(\mathbf{Q})^T))^{b(y_{ij})}}. \tag{11}
\end{aligned}
$$

The binomial likelihood has the follows:

$$\mathbf{Y}_i \sim \text{Binomial}(\varphi_{\mathbf{W}}(\mathbf{X}), \varphi_{\mathbf{W}}(\mathbf{Q}), \boldsymbol{\beta}, n),$$

$$\boldsymbol{\beta} = \boldsymbol{\beta}_X^T \boldsymbol{\beta}_Q, \text{ where } \boldsymbol{\beta}_X, \boldsymbol{\beta}_Q \sim \mathcal{N}(0, \Sigma_0),$$

$$\mathbf{X}_i \sim \mathcal{N}_D(0, I), \mathbf{Q}_j \sim \mathcal{N}_D(0, I),$$

$$L(y_j|\varphi_{\mathbf{W}}(\mathbf{X}), \varphi_{\mathbf{W}}(\mathbf{Q}), \boldsymbol{\beta}) = \prod_{i=1}^{N} C_{y_{ij}}^n \left(\frac{\exp(\varphi_{\mathbf{W}}(\mathbf{X})\boldsymbol{\beta}\varphi_{\mathbf{W}}(\mathbf{Q})^T)}{1 + \exp(\varphi_{\mathbf{W}}(\mathbf{X})\boldsymbol{\beta}\varphi_{\mathbf{W}}(\mathbf{Q})^T)}\right)^{y_{ij}}$$

$$\cdot \left(1 - \frac{\exp(\varphi_{\mathbf{W}}(\mathbf{X})\boldsymbol{\beta}\varphi_{\mathbf{W}}(\mathbf{Q})^T)}{1 + \exp(\varphi_{\mathbf{W}}(\mathbf{X})\boldsymbol{\beta}\varphi_{\mathbf{W}}(\mathbf{Q})^T)}\right)^{(n-y_{ij})}$$

$$= \prod_{i=1}^{N} C_{y_{ij}}^n \frac{(\exp(\varphi_{\mathbf{W}}(\mathbf{X})\boldsymbol{\beta}\varphi_{\mathbf{W}}(\mathbf{Q})^T))^{y_{ij}}}{(1 + \exp(\varphi_{\mathbf{W}}(\mathbf{X})\boldsymbol{\beta}\varphi_{\mathbf{W}}(\mathbf{Q})^T))^n}. \tag{12}$$

## B  Pseudo-code for ESS

---
**Algorithm 2** Elliptical Slice Sampler (Murray et al., 2010)

---
1: Choose an ellipse: $\boldsymbol{v} \sim \mathcal{N}_D(\mathbf{0}, \boldsymbol{\Sigma})$
2: Calculate log-likelihood threshold: $u \sim \text{Uniform}[0,1]$
$$\log y \leftarrow \log L(\boldsymbol{x}) + \log u$$
3: Draw an initial angle: $\theta \sim \text{Uniform}[0, 2\pi]$
   also defining a bracket: $[\theta_{\min}, \theta_{\max}] \leftarrow [\theta - 2\pi, \theta]$.
4: Calculate a new point, $\boldsymbol{x}' = \boldsymbol{x}\cos(\theta) + \boldsymbol{v}\sin(\theta)$.
5: **if** $L(\boldsymbol{x}') > \log y$ **then**
6:    accept the new point $\boldsymbol{x}'$;
7: **else**
8:    If $\theta < 0$ set $\theta_{\min} \leftarrow \theta$ otherwise $\theta_{\max} \leftarrow \theta$
9:    Draw $\theta \sim \text{Uniform}[\theta_{\min}, \theta_{\max}]$ and go back to steps 4.
10: **end if**

---

