# OpenReview forum: "Sparse data imputation with Bayesian non-linear factor analysis"
_TMLR — Rejected by TMLR_

### Review · Reviewer_zG8r · 2025-07-12

**Summary Of Contributions:**

The authors propose a non-linear factor analysis model that uses random Fourier features in two ways, and apply it for a missing data imputation problem. The method is evaluated on a collection of small synthetic problems and compared against a few closely related methods with positive results. The paper is specifically positioned only within the GPLVM family of methods, not considering other kinds of non-linear factor analysis methods or alternative techniques for missing value imputation.

**Audience:**

No

**Broader Impact Concerns:**

None.

**Claims And Evidence:**

No

**Requested Changes:**

The paper would need to be completely revised to be suitable for publication in any scientific venue. The most fundamental issues that would absolutely need to be fixed are:
1. Provide an actual justification for the method and tell a consistent narrative through the paper. As your model includes a non-parametric mixture component you need to explain its role already in the Introduction, you need to provide the mathematical details for the component on a sufficient level etc. Finally, Section 3 that proposes the method itself should typically be the longest one in a methodological paper, not the shortest!
2. Extend the related work to cover *at least* more recent works on factor analysis, other formulations for non-linear factor analysis, and other means for missing value imputation.
3. The experiments should be completely re-done. You should show somehow that the proposed inference scheme works well, likely by comparing it against trivial alternatives (e.g. just write the model in Stan/Pyro and run the default inference algorithms). To show it works well in the missing value imputation problem you should switch to concurrent missing value imputation benchmarks and evaluation metrics, and include broader range of comparison methods. It is important for the readers to get an insight on how well this fares in comparison to the currently used methods. You don't need to implement anything yourself, since there should be readily available implementations of the commonly used methods.

**Strengths And Weaknesses:**

**General overview:**
The paper addresses a common problem setup and provides a model variant that I believe is new. The work follows the standard structure of papers in the field and mostly satisfies and is written in clear language. However, it has several shortcomings as a scientific publication, from nearly non-existing motivation and explanation for the proposed model to poor empirical validation and lack of discussion and empirical comparison to other kinds of missing value imputation solutions. The presentation is also sub-standard for a scientific publication.

**On related work:**
The coverage of related work is in general sparse. Section 2 covers factor analysis methods solely using citations from the 20th century, with the exception of Mnih and Salakhutdinov (2007), omitting two decades of work in the field. Non-linear FA methods are covered only from the perspective of GPLVM, which is one stream of methods proposed for this but definitely not the only one. In particular, the vast literature on non-linear FA methods building on neural networks as the mapping from the latent variables to the outcomes is completely missing. Similarly, the paper completely omits all literature on missing value imputation despite considering it as the main application task. FA methods are one tool for missing value imputation, but a paper focusing on that would need to outline also the other tools for the same task. Overall, the paper is simply too detached from the concurrent work in the field to be of interest for people working in the field. As one concrete demonstration of this, there are in total only three references published during the last five years.

**Motivation and method:**
The core scientific contribution is poorly motivated and explained. The idea of learning latent representations for the features is only explained in the middle of Section 2.1, that is supposed to be background on GPLVMs, by suddenly switching from applications of GPLVM to this core aspect of the proposed work with a sentence starting "In some scenarios, we may also want ...". This is both a completely wrong place to explain this issue, which should be covered as a core topic already in Introduction, and an overall weak explanation of why it matters. Section 3 that proposed the new model is even more problematic, lacking **all** explanations. The authors simply write a massive equation that is a minor extension of Eq. (6), without providing any explanation for it. There are also other gaps in the story -- when Eq. (6) is introduced the authors suddenly start talking about non-parametric mixtures, without explaining them, even though the Introduction was completely focused on factor analysis and did not address cluster-like sub-structure in any way.

The posterior inference is standard but again poorly motivated within the literature. The authors falsely claim they *must* resort to MCMC even though other approximation schemes could also be used.  Given that the model is a standard probabilistic program it should not be too difficult to even provide an empirical comparison e.g. against stochastic-gradient variational approximation, but even if not explicitly doing this it would be important to avoid making false claims regarding MCMC being the only option. The specific MCMC sampler is reasonable but not justified either empirically or theoretically, and it would be useful to at least discuss some alternatives. For example, for the treatment of binary and binomial likelihoods the  Polya-gamma augmentation could be used as well, with possibly improved mixing over the slice sampler while still retaining analytic Gibbs conditionals.

The authors do a good job at explaining the standard RFF trick, but the paper is not self-contained in terms of the mathematical description, especially lacking in treatment of non-parametric components.

**Experiments:**
The experiments are reasonable, but not particularly interesting for the community and fall severely short of how missing value imputation methods are today evaluated. With the exception of the artificial experiment on modified CIFAR-10, all of the data sets are the classic benchmarks from approximately 20 years ago, and by today's standards they are small and non-representative of the challenges of modern data analysis. There are dedicated missing data imputation benchmarks, like DACMI. Why not use them?

The comparison methods are limited to the closest methodological neighbours, ignoring broad literature of concurrent missing value imputation methods and even most of non-linear factor analysis methods. It is important to show the difference to PPCA and GPLVM to illustrate the importance of the new components, but the experimentation is not sufficient for establishing whether the method is competitive in this task. The current state-of-the-art for missing value imputation is extremely far from methods like PPCA, typically using ensembles that leverage (generative) deep learning techniques, they support also problems where the entries are not missing at random, and their performance is evaluated on dedicated benchmarks with appropriate evaluation metrics. All of these perspectives are missing from this work.

The presentation of the experimental section is also problematic. Since all experiments are of essentially the same nature, there is no need to split the experiments based on the data set -- you could just say that you use N standard benchmark data sets and present the results in one go. The results are presented only using MSE with no additional insights on how the methods work -- except for the down-stream analysis task on MovieLens -- for example never showing that the sampler works correctly or evaluating its mixing or computational efficiency. Even questionable results (e.g. MSE for Poisson RFLVM and PPCA is way bigger than the rating scale in Table 6) are left unexplained. Finally, the table formatting is inconsistent (font size), overly crowded with lines, and in general not professional in appearance.

---

### Review · Reviewer_5vxp · 2025-07-20

**Summary Of Contributions:**

The paper proposes an approach that performs non-linear modeling of the latent variables and factors in a factor analysis model. This is achieved by using random Fourier features to approximate Gaussian processes. The paper develops an MCMC-based procedure to sample from the proposed model. Experimental results are reported for the data imputation task, with a demonstration of outperformance compared to other latent variable methods.

**Audience:**

Yes

**Claims And Evidence:**

No

**Requested Changes:**

C1. Please add a discussion of (at least some of) the most relevant recent work on data imputation and explain why the techniques are or are not included as baseline comparisons.

C2. The current choice of baselines needs much better justification, and I would encourage the authors to expand their comparison to include other recently proposed methods. PPCA is an old method, so while it should be included as a baseline, it can’t be considered state-of-the-art. It’s not ideal to then compare only to two other techniques that are very similar to the introduced method. In order to support the claim that “our proposed model is effective for missing data imputation, especially when the percentage of missing data is high”, there needs to be a clearer indication of what other methods are capable of achieving.

C3. The main model is presented in equation (7), but there are only two sentences that discuss this model. The paper should provide at least a full paragraph to discuss the main model in the paper.

C4. Aside from Section 4.5, the analysis of experimental results is superficial, with only MSE tables presented. The paper should include a more meaningful analysis of the results. Some aspects of interest include the sampling behaviour and clearer insights into the nature of the factors that are being captured by the model.
Minor point: It’s not clear how (4) follows from (3) by “Monte Carlo integration”.

**Strengths And Weaknesses:**

Strengths

S1. The paper develops a novel latent variable model that introduces greater flexibility in capturing non-linear aspects of the data.

S2. The paper provides a detailed procedure for sampling from the proposed model.

S3. The model is tested on multiple datasets and compared to closely-related techniques.

Weaknesses

W1. The paper is a relatively minor extension of the Random Feature Latent Variable Model, with the main change being that both the latent factors and latent variables are modelled using Gaussian processes. Although this does introduce some technical challenges in sampling, the proposal is somewhat incremental.

W2. The discussion of related work is very limited. The model is presented as a tool for data imputation, but there is no discussion of recent methods for performing data imputation.

W3. The paper does not really discuss the choice of baseline methods. Moreover, the technique is only compared to a couple of recent closely related methods that don’t clearly represent the state-of-the-art. If the case is being made that this is a useful approach for data imputation, there should be a comparison to multiple recent techniques (or at least an explanation as to why such a comparison is not meaningful or relevant).

W4. The analysis of the experimental results is limited. For the most part, there is just a presentation of tables of MSE. The paper would be considerably more interesting if there were a characterization of the sampling behaviour, including diagnostics, and examination of the types of factors and weightings that are being constructed.

W5. There are multiple grammatical and English usage errors. In several cases, they make it difficult to understand the presented technical material. The authors should run a grammar checker and conduct a careful proof-reading to ensure that most errors are eliminated.

---

### Review · Reviewer_CQj8 · 2025-08-01

**Summary Of Contributions:**

The authors propose a new imputation method based on non-linear factor models with random Fourier features as a means to approximate a Gaussian process for the latent factors and loadings. This is done by building upon work for random feature latent variable models (RFLVMs), but using separate latent variables for features and observations to produce a random feature latent factor analysis (RFLA) model, which leverages an elliptical slice sampler for Bayesian analysis. Experiments on real-world datasets indicate that the proposed approach is effective, especially in high-missingness scenarios.

**Audience:**

Yes

**Broader Impact Concerns:**

No concerns regarding ethical implications.

**Claims And Evidence:**

No

**Requested Changes:**

Major:
- The methods are poorly presented. The approach in (7) is not justified, especially relative to (6) and how it makes it suitable for imputation. Notation is poorly addressed and many details about the generative model and simulation framework are taken for granted, thus not making the methods accessible to someone not deeply familiar with Bayesian analysis and GPLVMs.
- The experiments are largely underwhelming:
  - There are very little details about the data, simulation settings, data partition, and evaluation.
  - How is the data processed?
  - What are the hyperparameter values for all models?
  - How is the simulation run and how is convergence monitored/assessed?
  - How many samples are used for simulation and inference?
  - Why are GPLVM and RDLVM worse than PPCA provided that arguably for are initialized from PPCA?
  - How are discrete/categorical variables handled and why use MSE for those?
  - Why are the results for Binomial RFLFA better than Poisson RFLFA and so close to the much simpler and misspecified PPCA?
- Consider an ablation study to explore whether the imputation gains are due to the new formulation or the simulation strategy based on elliptical sampling.
- The authors consider experiments where data is missing completely at random, however, in many practical scenarios including gene expression, that is not the case. Why not consider more realistic scenarios?
- It is not clear what is the rationale of using a model like the one proposed to impute pixels on images and if there is one, why not consider models specifically developed for imputation/inpainting? Moreover, why not compare in general with methods specifically developed for imputation?
- As presented it will be difficult for readers to implement and reproduce the experiments in the paper.

Minor:
- Some quantities are not defined: {\cal N}(,), \sigma, I, \beta, Ga(,), NIW(,,,,), a_\alpha, b_\alpha, \mu_0, \nu_0, \gamma_0, \Psi_0, n_k^-m, n_k, \upsilon_k (which seems to be a typo), \upsilon_0, \pi_\eta, \eta
- There is a missing [ in (3).
- (6) abuses notation by implying that one can sample from a likelihood function.
- There are also several typos in the text and mathematical notation that make it difficult to follow the paper.
- What are one Poisson, two Poisson, one binomial and two binomial in Table 4?
- Details of the methods in Table 7 are missing.

**Strengths And Weaknesses:**

Considering the way the methodology and results are presented, it is difficult to pinpoint the strengths of the submission other than the general value of accurate imputation methods, especially in very sparse datasets and collaborative filtering settings like the one presented in the MovieLens experiment.

For details on the weaknesses of the submission, see the requested changes below.

---

> ### Author Response · Authors · 2025-08-04
> **We will clarify methods, expand experiments, and address all identified issues.**
>
> We sincerely thank you for your thorough evaluation and constructive feedback. We are grateful for the opportunity to improve our manuscript, and we address each of your comments in detail below. All the suggested changes and clarifications will be incorporated into the revised version of the paper.
>
> We agree that the model in Eq. (7) was insufficiently justified. To address this, we will provide a more consistent narrative throughout the paper. Specifically, we will dedicate a full paragraph to describing and discussing the main model, including an explanation of the dual latent variable structure. Additionally, we will restructure the model section to clearly explain each component, making it more accessible to all readers.
>
> We will significantly expand the experimental section to include detailed descriptions of each dataset, general preprocessing steps, and missingness scenarios, along with full hyperparameter configurations for all models.
> We performed 2000 iterations of sampling for inference and will further analyze the sampling behavior by providing trace plots.
> We will adopt more suitable evaluation metrics for categorical data, such as in the case of the spam dataset.
> In our experiments, although all models are initialized from PPCA, the non-convexity of these models, combined with poor convergence or overfitting in sparse settings, can sometimes result in worse performance compared to PPCA. The Binomial RFLFA performs better might because of the compatibility between the true data and the assumed likelihood. Poisson RFLFA tends to underperform in binary data due to misspecification. The closeness to PPCA arises in low-sparsity regimes or when the non-linear structure is weak.
>
> We appreciate the suggestion to include an ablation study. The comparison between RFLFA and RFLVM is conducted both using elliptical slice sampling (ESS), as the previously used optimization method cannot handle missing data. In the revised version, we will include results comparing Polya-Gamma augmentation and variational inference to further analyze the impact of different inference strategies.
>
> We agree that the MCAR assumption is unrealistic in many real-world applications, but it remains a useful setting for evaluating the imputation performance of models. We will consider new experiments using missing not at random (MNAR) mechanism.
>
> We wish to focus on comparing our model with those in the GPLVM family due to their similar nature and latent variable structure. Given the structure of our model, it does not perform well enough on image inpainting tasks to compete with deep neural network-based methods. But we will expand our discussion of missing data imputation methods and include comparisons with other commonly used methods.
>
> We will prepare a fully documented, publicly available GitHub repository that includes all scripts, data, and instructions to reproduce the experiments.
>
> Thanks for pointing out the minor issues and typos in the manuscript. We will carefully revise the paper to address all of them. “One Poisson, two Poisson, one binomial and two binomial” refer to whether the model uses one or two sets of latent variables for rows and columns. We will rename them to ensure consistency with the model names used elsewhere in the paper.
>
> Again, we thank you for the insightful and detailed comments. These suggestions will help us to improve the quality of our work. We hope the revised version will meet the expectations set forth in the review.

---

### Decision · Action_Editor_hQoa · 2025-09-04

**Recommendation:** Reject

**Audience:**

No

**Audience Explanation:**

The paper will require substantial work to be suitable for publication and interesting for the community. Many criticisms were raised by all three reviewers and the authors did not provide a detailed response or make changes to the paper.

**Claims And Evidence:**

No

**Claims Explanation:**

The claims are not properly supported by theory or empirical evidence. For example, the authors claim the performance is comparable to competing missing value imputation methods, but the paper does not include any state-of-the-art imputation methods as comparisons and the empirical evaluation is done only on simplified problems that would not even reveal differences between the methods well.